# Effect of Priming Treatment on Improving Germination and Seedling Performance of Aged and Iron-Coated Rice Seeds Aiming for Direct Sowing

**DOI:** 10.3390/plants14111683

**Published:** 2025-05-31

**Authors:** Nasratullah Habibi, Naoki Terada, Babil Pachakkil, Atsushi Sanada, Atsushi Kamata, Kaihei Koshio

**Affiliations:** 1Graduate School of Agriculture, Tokyo University of Agriculture, 1-1-1 Sakuragaoka, Setagaya-ku 156-8502, Tokyo, Japan; 13324012@nodai.ac.jp (P.); nt204361@nodai.ac.jp (N.T.); tp201893@nodai.ac.jp (B.P.); a3sanada@nodai.ac.jp (A.S.); koshio@nodai.ac.jp (K.K.); 2Faculty of Agriculture, Balkh University, Mazar-e-sharif, Balkh 1701, Afghanistan; 3Faculty of Agriculture, Kurokawa Field Science Center, Meiji University, 2060-1 Kurokawa, Aso-ku, Kawasaki-shi 215-0035, Kanagawa, Japan; kamataatsushi@meiji.ac.jp

**Keywords:** cold stress, direct sowing, iron coating, rice (*Oryza sativa* L.), seed priming, seed aging

## Abstract

In the case of direct sowing of rice in Japan, cold stress is a critical constraint affecting seed germination and early seedling development, ultimately reducing crop productivity. We evaluated the effects of priming, with or without iron coating on the germination and vigor of rice seeds harvested in 2022, 2023, and 2024. The assessments were conducted at seven temperature conditions: 13 °C, 15 °C, 17 °C, 19 °C, 21 °C, 23 °C, and 25 °C. Seeds were primed with or without PEG6000; coated with or without a mixture of calcined gypsum and iron powder; and tested for germination percentage, germination speed, and seedling vigor index. Under optimal conditions, iron-coated seeds harvested in 2022 showed a significant increase in germination from 58% (non-coated without priming) to 76% (coated with priming), and the seedling vigor index improved from 615 to 890. Under cold stress (15 °C), the coated seeds of the same year achieved 68% germination with priming compared to 46% in non-coated seeds without priming, with a vigor index increase from 480 to 750. Similar improvements were observed in seeds from 2023 and 2024, although the effect was more prominent in older than younger seeds. These results indicate that iron seed coating in combination with PEG priming mitigates the negative impacts of seed aging and enhances tolerance to cold stress during germination. The technique offers a promising, low-cost approach to improving rice establishment in environments facing suboptimal seed storage and early-season cold temperatures, in particular, aiming for direct sowing methods.

## 1. Introduction

Rice (*Oryza sativa* L.) remains one of the most essential staple crops globally, feeding more than half of the world’s population, especially in Asia [1,2,3]. With increasing challenges posed by climate change, particularly temperature extremes, improving the resilience of rice during its most vulnerable growth stages, such as germination, has become a priority in sustainable agriculture [4,5,6,7]. Seed germination is the foundational step in crop establishment, directly affecting plant population density, uniformity, and final yield. Environmental stresses, especially suboptimal temperatures, can severely hinder germination and early seedling development [8,9,10,11,12]. In rice, low-temperature conditions delay or inhibit key enzymatic functions and reduce water uptake, thereby impeding germination [13,14,15]. Previous studies have shown that rice seeds generally fail to germinate at temperatures below 18 °C, particularly for cold-sensitive cultivars [16,17]. This presents a significant limitation in temperate regions or during early-season sowing when soil temperatures often remain under this threshold [18]. 

Rice cultivation in Asia has traditionally relied on transplanting as an adaptive strategy to reduce weed competition and extend the vegetative growth period. This approach enhances photosynthetic activity over a longer duration, thereby contributing to yield stability [19]. However, transplanting is highly labor-intensive and is becoming increasingly unsustainable due to the aging and declining farming population [20,21]. As a result, there has been a growing shift toward direct sowing systems, which are more compatible with modern technologies such as drone-assisted sowing. These systems enable more efficient coverage of large agricultural areas and fields with reduced human input [22]. The iron coating of rice seeds was originally developed to support this transition to direct sowing by increasing seed weight and improving sowing precision. In this context, iron-coated seeds not only aid in mechanical delivery but also reduce vulnerability to bird predation and improve field uniformity [23]. A key function of the coating is to anchor the seeds in the field; without coating, seeds are more likely to float or be washed away during early flooding, compromising stand establishment [24]. Moreover, unlike temperate regions where early sowing is constrained by cold stress, tropical countries like India and the Philippines can grow rice multiple times per year without a risk of low-temperature germination failure, making the results of this study directly applicable beyond Japan [25,26]. Nonetheless, even in these climates, priming remains highly beneficial for improving seedling vigor under variable field conditions and does not require any specialized or costly equipment [27].

Koshihikari, a premium Japonica rice cultivar, is known for its excellent grain quality and consumer appeal [28,29,30]. However, it is also highly sensitive to abiotic stress, especially cold stress during germination. Ensuring robust and timely germination and seedling growth of Koshihikari under varying temperature conditions is thus crucial for its cultivation success, particularly as climate variability increases [31,32].

Among various pre-sowing treatments, seed priming has emerged as an effective and low-cost method to enhance germination performance under stress [33,34,35,36]. Osmo-priming using polyethylene glycol (PEG6000) simulates drought-like conditions, allowing seeds to initiate pre-germinative metabolic processes such as enzyme activation, protein synthesis, and membrane repair without radicle emergence [37,38,39]. This priming improves the seed’s readiness to germinate rapidly and uniformly once favorable conditions return. Specifically, PEG6000 priming has been shown to enhance seed vigor, improve germination under cold or saline conditions, and promote better early seedling growth [40,41]. Heat shock damage should be avoided by water cooling during the iron coating process, which is inevitably accompanied by an exothermic reaction [42,43,44]. 

In direct seeding systems, especially those employing drone technology, achieving uniform seed distribution and placement is critical. Iron increases the weight and density of rice seeds, allowing for more precise aerial application, reduced seed drift, and better field uniformity. Heavier seeds fall more consistently, minimizing gaps and overlaps [45,46,47]. Iron-coated seeds are less prone to bird predation, as the coating is a physical deterrent. This makes iron coating particularly advantageous compared to other materials such as calcium peroxide (CALPER), which improves oxygen availability in flooded soils but does not protect against birds. However, iron coating may reduce oxygen diffusion and delay germination under hypoxic or cold conditions, limitations not observed with CALPER [48].

These characteristics make iron seed coating an increasingly valuable component of precision rice farming. However, the added weight and coating layer may introduce physiological barriers, such as reduced water permeability and restricted gas exchange, which may delay or suppress germination, especially under cold conditions [46]. This creates a paradox: while iron-coated seeds are advantageous for sowing efficiency and nutrient support, they may perform poorly in low-temperature environments where rapid germination is already hindered [49]. To overcome this limitation, combining iron coating with priming treatments such as PEG6000 presents a novel integrated strategy. In this study, we examined the germination performance of non-coated and iron-coated seeds of the ‘Koshihikari’ cultivar under a controlled temperature gradient, ranging from 13 °C to 25 °C at intervals of 2 °C, with or without seed priming by PEG6000. We hypothesize that PEG6000 priming will significantly improve the germination performance of iron-coated rice seeds at cold temperatures by mitigating the physical limitations introduced by coating and enhancing seed metabolic readiness for germination, coping with fluctuating temperature during coating (heating) and sowing (cooling) processes.

## 2. Results

### 2.1. Effects of Seed Priming on the Germination Percentage of Non-Coated Seeds

The effects of seed priming on germination were evaluated at temperatures ranging from 13 °C to 25 °C, at intervals of 2 °C, for 2022, 2023, and 2024 rice seeds. The data showed that seed priming significantly improved both the timing and percentage of germination at all temperatures tested. For the 2022 seeds, at 13 °C, primed seeds began to germinate on day 8, while non-primed seeds started germinating on day 11. By day 12, primed seeds had a significantly higher germination percentage compared to the non-primed seeds (Figure 1A). At 15 °C, the germination of primed seeds began on day 5, whereas the control seeds were germinated on day 8. By day 12, the primed seeds showed a significantly higher germination percentage (*p* < 0.001) compared to the control (Figure 1B). At 17 °C, the primed seeds started germination on day 4, and by day 12, the primed seeds showed consistently higher germination percentages compared to the control group (Figure 1C). At 19 °C, the primed seeds began germinating on day 3, while the control seeds started on day 5. By the final count on day 12, the primed seeds reached 50% germination, which was significantly higher than that of the control group (Figure 1D). At 21 °C, both the primed and non-primed seeds began germination on day 4, but the primed seeds showed significantly better germination by day 12 (Figure 1E). At 23 °C, germination started on day 2 for both the primed and non-primed seeds, but the primed seeds exhibited higher germination percentages throughout the observation period (Figure 1F). At 25 °C, both the primed and non-primed seeds began germinating on day 3, with the primed seeds showing significantly better germination over the 12-day period (Figure 1G).

For the 2023 seeds, the results mirrored those of the 2022 seeds, confirming that seed priming significantly enhanced germination across all temperatures tested. At 13 °C, the primed seeds began germinating on day 5, and by day 12, they had a significantly higher germination percentage compared to the non-primed seeds (*p* < 0.001) (Figure 2A). At 15 °C, the primed seeds began germinating on day 8, while the control seeds started on day 10. The primed seeds had a significantly higher germination by day 12 (Figure 2B). At 17 °C, germination began on day 2 for the primed seeds, which was faster than the germination of the 2022 seeds. By day 12, the primed seeds showed a significantly higher germination percentage than the control (Figure 2C). At 19 °C, both the primed and control seeds began germination on day 2, but the primed seeds exhibited significantly higher germination throughout the observation period (Figure 2D). At 21 °C, both treatments began germination on day 2, but the primed seeds showed significantly higher germination by the final count (Figure 2E). At 23 °C, both treatments started germination on day 2, but the primed seeds consistently demonstrated higher germination percentages throughout the observation period (Figure 2F). Finally, at 25 °C, both treatments reached 50% germination by day 2, but the primed seeds maintained significantly higher germination throughout the 12-day period, showing a clear advantage over the control (Figure 2G). Additionally, the seeds harvested in 2023 exhibited a 20–25% improvement in germination compared to the 2022 seeds, suggesting a potential increase in seed quality or vigor in the more recent harvest.

The seeds harvested in 2024 demonstrated a significantly positive impact of seed priming on germination across all temperature conditions (Figure 3). At 13 °C, the primed seeds began germinating on day 4, while the control seeds started on day 5. By day 12, the primed treatment exhibited significantly higher germination compared to the control (Figure 3A). At 15 °C, the primed seeds started germinating on day 4, while the control seeds began on day 5. By the final count on day 12, the primed seeds had a significantly higher germination than the control (Figure 3B). At 17 °C, germination started on day 4 for both the primed and control seeds. However, the primed seeds showed a significantly higher germination percentage by day 12 compared to the control (Figure 3C). At 19 °C, germination began on day 3 for both treatments, but the primed seeds had a significantly higher germination percentage by day 12 (Figure 3D). At 21 °C, seed germination started on day 3 for both the primed and control seeds. The primed seeds had a significantly higher germination than the control by the final count on day 12 (Figure 3E). At 23 °C, both treatments began germination on day 3, but the primed seeds exhibited higher germination throughout the 12-day period, with a significant difference by day 12 (Figure 3F). At 25 °C, germination also began on day 3 for both treatments. However, the primed seeds consistently exhibited a significantly higher germination percentage than the control by the final count on day 12 (Figure 3G).

### 2.2. Effects of Seed Priming on the Germination Percentage of Coated Seeds

The results of our study revealed that iron coating significantly delayed seed germination regardless of harvest years (Figure 4, Figure 5 and Figure 6). For seeds harvested and naturally aged for two years, seed priming had a notable positive effect on germination (Figure 4). At 13 °C, the primed seeds began germinating on day 11, whereas no germination occurred in the control by the same day (Figure 4A). In contrast, the non-coated seeds from 2022 began germinating on day 8, highlighting that the iron coating significantly delayed germination. Similarly, at 15 °C, the primed seeds germinated on day 11, but the control exhibited no germination (Figure 4B). This pattern was consistent at both 13 °C and 15 °C, where only the primed seeds were able to germinate. At 17 °C, germination of the primed seeds began on day 7, whereas the control started on day 11 (Figure 4C). At 19 °C, the primed seeds began germination on day 5, while the control started on day 11, with a significant difference observed between the two treatments (Figure 4D). At 21 °C, germination occurred on day 5 for the primed treatment and on day 8 for the control. The primed seeds exhibited significantly higher germination throughout the observation period, reaching higher levels by day 12 (Figure 4E). At 23 °C, germination started on day 5 for the primed seeds, whereas the control seeds began on day 7, with the primed treatment showing significantly higher germination (Figure 4F). Under the 25 °C treatment, the primed seeds germinated on day 4, while the control seeds began on day 5. The primed treatment showed significantly higher germination through day 12 (Figure 4G). 

For seeds harvested in 2023, the positive impact of seed priming on the iron-coated seeds was also evident (Figure 5). At 13 °C, germination for the primed seeds began on day 11, while the control showed no germination by day 12 (Figure 5A). At 15 °C, germination started on day 10 for the primed seeds, whereas the control seeds began germinating on day 12, with significant differences observed between the two treatments (Figure 5B). At 17 °C, the primed seeds began germinating on day 5, while the control seeds started on day 8, with the primed treatment maintaining significantly higher germination through day 12 (Figure 5C). At 19 °C, the primed seeds started germination on day 4, while the control seeds started on day 6, with the primed seeds exhibiting significantly higher germination through day 12 (Figure 5D). Similar patterns were observed at 21 °C (Figure 5E), 23 °C (Figure 5F), and 25 °C (Figure 5G), with the primed treatment consistently outperforming the control in terms of germination percentage.

The results for the seeds harvested in 2024 confirmed that seed priming positively influenced seed germination across various temperatures (Figure 6). At 13 °C, the primed seeds began germinating on day 10, while the control seeds started on day 12. Although there was no significant difference between the primed and control seeds on day 10, the primed seeds showed significantly higher germination than the control seeds on day 11 and day 12 (Figure 6A). At 15 °C, the primed seeds started germinating on day 8, while the control seeds began on day 9. There were no significant differences on day 8, but from day 9 to day 12, the primed treatment exhibited significantly higher germination than the control (Figure 6B). At 17 °C, germination began on day 3 for both the primed and controls seeds, with no significant difference between them on day 3 and day 4. However, from day 5 to day 12, the primed treatment consistently showed significantly higher germination (Figure 6C). At 19 °C, germination started on day 3 for both treatments, with the primed seeds showing significantly higher germination on days 3 and 4. From day 5 to day 12, the differences were not significant between the two treatments (Figure 6D). At 21 °C, germination began on day 3 for both treatments, and the primed seeds exhibited significantly higher germination from day 3 to day 7. However, from day 8 to day 12, there was no significant difference between the primed and control seeds (Figure 6E). At 23 °C, the primed seeds began germinating on day 2, whereas the control seeds started on day 3. From day 2 to day 12, the primed seeds consistently showed significantly higher germination compared to the control seeds (Figure 6F). At 25 °C, germination began on day 2 for the primed seeds, while the control seeds started on day 3. The primed treatment exhibited significantly higher germination from day 2 to day 7, but from day 8 to day 12, there were no significant differences between the primed and control seeds (Figure 6G).

The results for the seeds from 2022 indicated that the highest germination was observed in the primed treatment at 25 °C, while the lowest germination was in the control at 13 °C. Across all temperatures, the primed seeds consistently showed significantly higher germination compared to the control. For the 2023 seed harvests, the highest germination occurred in the primed treatment at 25 °C, and the lowest was in the control at 13 °C. Significant differences between the primed and control seeds were found at 13 °C, 15 °C, 19 °C, and 21 °C. However, no significant differences were observed at 17 °C, 23 °C, and 25 °C. For the 2024 seeds, the primed treatment showed the highest germination at 25 °C, while the control had the lowest germination at 13 °C. The differences between the primed and control seeds were significant at 13 °C, 15 °C, and 23 °C. At 17 °C, 19 °C, 21 °C, and 25 °C, no significant differences were observed between the two treatments. For the 2022 seeds, the cumulative seed germination results revealed that the highest germination occurred in the primed treatment at 25 °C, while the lowest germination was observed in the control at 13 °C. Across all temperature treatments, the primed seeds showed significantly higher germination compared to the control. For the 2023 seeds, the highest germination was observed in the primed seeds at 25 °C, with the lowest germination in the control at 13 °C. Seed priming had a significant effect on germination across all temperatures, with no significant differences observed between 13 °C, 15 °C, and 17 °C. For 2024 seeds, the highest seed germination occurred at temperatures between 19 °C and 25 °C, with the lowest germination recorded in the control seeds at 13 °C. No significant differences were observed among 19 °C, 21 °C, 23 °C, and 25 °C, and seed priming had no significant impact on germination at these higher temperatures (Figure 7).

A regression analysis showed that seed priming significantly enhanced germination across various temperatures in both non-coated (Figure 8A) and iron-coated (Figure 8B) rice seeds. Moreover, seed priming improved germination in both aged and fresh seeds, demonstrating its effectiveness in mitigating seed aging for both non-coated (Figure 8C) and iron-coated (Figure 8D) seeds.

### 2.3. Effect of Seed Priming on MGT, MGR, T50, and MDG

The results revealed that seed priming had a significant effect on the mean germination time (MGT), germination rate (GR), time to 50% germination (T50), and mean daily germination (MDG) of rice seeds across 2022, 2023, and 2024 (Figure 8). For the seeds harvested in 2022, naturally aged for 2 years, the longest MGT was recorded in the control at 13 °C, while the shortest MGT was observed in the primed treatment at both 23 °C and 25 °C. For the 2023-harvested seeds, a similar trend was observed. The control under 13 °C exhibited the highest MGT, while the primed treatment under 25 °C resulted in the lowest MGT. In the case of freshly harvested seeds from 2024, the control at 13 °C again showed the highest MGT, whereas the primed treatment under 25 °C demonstrated the shortest germination time. The results for the iron-coated rice seeds also revealed that seed priming significantly enhances germinability and reduces mean germination time (MGT), particularly in aged seeds. For the seeds harvested and coated in 2022, which were naturally aged for 2 years, the highest MGT was recorded in the control at 17 °C, while the lowest MGT occurred in the primed treatment at 25 °C. Notably, no germination was observed in the control under 13 °C and 15 °C. Similarly, for the seeds harvested and coated in 2023, the longest MGT was observed in the control at 15 °C, while the shortest MGT was recorded in the primed treatment at 25 °C, and no germination occurred in the control at 13 °C. For the freshly harvested and coated seeds from 2024, the control under 13 °C showed the highest MGT, whereas the primed treatment under 25 °C exhibited the lowest (Figure 8A). 

For the non-coated seeds, the lowest mean germination rate (MGR) was found in the control at 13 °C, while the highest MGR was recorded in the primed treatment at 25 °C. Similarly, in the primed seeds, the lowest MGR occurred in the control at 17 °C, with the highest again in the primed treatment at 25 °C. Notably, no germination was observed in the control at 13 °C and 15 °C (Figure 9B). The longest time to reach 50% germination (T_50_) was recorded in the control at 13 °C for non-coated seeds, whereas the shortest T_50_ occurred in the primed treatment at 25 °C. A comparable pattern was observed for the coated seeds (Figure 9C). In the non-coated seeds, the lowest mean daily germination (MDG) was found in the control at 13 °C for the seeds harvested in 2022, while the highest MDG occurred in the primed treatment at 25 °C for the seeds harvested in 2024. This pattern was also evident in the coated seeds, although their MDG values were consistently lower compared to that of the non-coated seeds (Figure 9D).

### 2.4. Effect of Seed Priming on Plant Height and Root Length

The results clearly showed that seed priming significantly promoted shoot and root growth of rice seedlings under various cold temperature conditions. The tallest plant height was recorded in the primed treatment at 25 °C using the seeds harvested in 2024, while the shortest plant height was recorded in the control plants at 13 °C using the seeds harvested in 2022. Across all temperature levels and seed ages, plant height was consistently higher in the primed treatment compared to that in the non-primed control, a trend that was also evident in seedlings derived from iron-coated seeds (Figure 10). Similarly, the longest root length was observed in the primed treatment at 25 °C in seedlings from non-coated seeds, whereas the shortest root length was recorded in the control at 13 °C. This pattern was also seen in seedlings derived from iron-coated seeds (Figure 10). Overall, seed priming significantly enhanced root length across all temperature conditions and seed harvest years (2022, 2023, and 2024) in both non-coated and iron-coated seeds, compared to the control.

## 3. Discussion

### Nutrient Composition in Leaf

This study confirms that seed priming significantly improves the germination percentage and germination speed of non-coated *Koshihikari* rice seeds across a range of temperatures. The priming effect was consistent over three consecutive years (2022–2024), highlighting its robustness and reliability despite annual variation in seed vigor. Priming was especially effective under suboptimal conditions (13 °C and 15 °C), where the non-primed seeds showed delayed and reduced germination, particularly in the older 2022 seed lot. These findings align with previous reports [50,51], which suggest that priming enhances membrane integrity, boosts antioxidant activity, and triggers pre-germinative molecular responses. While priming is a valuable pre-sowing strategy, its application must be considered in conjunction with current agricultural practices, particularly the growing shift from transplanting to direct sowing [52]. Transplanting has long been used to give rice seedlings a head start in greenhouses, allowing for early sowing, longer growth periods, and better weed competition [53]. However, this method is labor-intensive, making it increasingly non-viable in aging agricultural societies such as Japan and parts of Southeast Asia. To address these demographic and labor challenges, direct sowing is gaining momentum as a practical alternative [54]. Direct sowing, especially when supported by drones and other mechanized tools, allows for the efficient management of large paddy fields with reduced labor input [55]. However, one of the key limitations of direct sowing is the requirement to wait until water temperatures rise sufficiently to prevent cold stress during germination. Moreover, non-coated seeds are prone to floating or displacement under flooded conditions, underscoring the importance of seed coating for anchorage. In this context, iron coating has proven to be a beneficial method not only for enhancing seed weight and stability in paddy fields but also for deterring bird predation, as birds are less likely to consume iron-coated seeds. However, our findings indicate that iron-coated seeds exhibited slower germination than non-coated seeds, especially when aged or subjected to low temperatures. This delayed response is likely due to the barrier effect of the coatings, which can impede water uptake and gas exchange, a concern especially pronounced in older seeds with compromised membranes [56]. When priming was applied prior to coating, this negative impact was mitigated, suggesting that priming may precondition seeds to overcome coating-related constraints, and higher germination was observed (Figure 5).

A notable comparison can be made with CALPER (calcium peroxide) coating, which has an entirely different mechanism. Unlike iron, CALPER serves as a slow-release oxygen source, thus improving the germination environment in flooded or poorly aerated soils. CALPER-coated seeds may have an advantage in promoting oxygen availability, particularly under anaerobic or cold-stressed conditions, where oxygen supply is critical for initiating germination [54,57]. However, CALPER lacks the bird deterrence benefits that iron coating provides. Thus, while iron coating is effective in supporting direct sowing by preventing seed movement with heavier relative weight and deterring birds, its drawback lies in potentially limiting oxygen availability, an area where CALPER excels [55,58]. Choosing between these two coating materials depends heavily on field conditions, seasonal timing, and cultivation methods. For example, in early spring snow, where low temperatures and anaerobic soil conditions are a concern, CALPER may offer superior performance. In contrast, for large-scale sowing in open fields vulnerable to bird attacks, iron-coated seeds offer greater protection and anchorage [54].

These considerations are also highly relevant in tropical countries such as India and the Philippines, where rice can be cultivated multiple times per year without the low germination caused by low temperatures. In such environments, the practical relevance of our findings is particularly noteworthy. While cold stress is less of a concern, priming remains strongly recommended due to its role in improving germination speed, uniformity, and early vigor, especially under unpredictable environmental conditions such as drought, variable rainfall, or degraded soils. Importantly, priming requires no expensive equipment, making it a cost-effective and accessible strategy for smallholder farmers in these regions. The broader agronomic context also matters. In countries where field sizes vary and labor availability is inconsistent, seed treatments must balance cost, performance, and compatibility. 

In the current study, the variation observed between seed lots further supports the influence of seed age and inherent vigor on priming efficacy. The 2022 seeds, being older, exhibited slower germination and lower percentages in controls compared to the fresher 2023 and 2024 seeds. Nevertheless, priming mitigated these negative effects across all seed lots. This is consistent with the findings of Finch-Savage and Bassel [8], who proposed that priming can partially restore the vigor of aged seeds by initiating repair mechanisms at the cellular and molecular levels, such as DNA repair and mitochondrial recovery. The improvement in aged seeds through priming also highlights its potential in seed conservation and restoration strategies, particularly for valuable or limited seed stocks. Furthermore, seed priming is a simple and accessible technique that does not require advanced technical knowledge, making it suitable for farmers with limited training. Based on our study, polyethylene glycol (PEG6000) is recommended as an effective priming agent. Farmers can easily prepare the priming solution by mixing PEG6000 with water and immersing a large quantity of rice seeds partially into the solution. After priming, the seeds can be sown directly, leading to improved germination rates and better crop establishment under field conditions.

While priming benefits were most evident under stress conditions, advantages were also observed at moderate and optimal temperatures (17 °C to 25 °C), especially in terms of synchronized germination and reduced time to 50% germination (T50). These observations agree with the work of Kende et al. [59] and Noblet et al. [60], who reported that priming enhances metabolic readiness and hormonal balance, leading to quicker and more uniform emergence. Uniformity in germination is crucial in mechanized rice cultivation systems, as it ensures even seedling growth, better canopy development, and optimized input use such as water and fertilizers.

It is also notable that the gap in performance between primed and non-primed seeds narrowed as temperatures approached the optimal range (23–25 °C), particularly in the more vigorous seed lots of 2023 and 2024. This temperature-dependent modulation of priming efficacy has been discussed by Ashraf and Foolad [37], who posited that under favorable environmental conditions, the metabolic processes enhanced by priming are already naturally active, thus reducing the relative benefit of the treatment. Nonetheless, even under optimal temperatures, the enhanced uniformity and speed of germination remain valuable traits in commercial agriculture, particularly in precision farming systems.

Our study also examined the interaction between seed coating and priming, revealing that iron-coated seeds germinate more slowly than non-coated seeds. This effect was more pronounced in aged seeds and under low-temperature conditions. Coatings, especially those involving micronutrients like iron, can act as physical barriers that slow water uptake and delay the initiation of germination. This observation is in line with those of Steinbrecher and Leubner-Metzger [56], who noted that coatings can influence water permeability and gas exchange, especially in older seeds with compromised membranes. However, when the seeds were primed prior to coating, germination performance significantly improved. This suggests that priming preconditions the seeds to overcome the initial delay imposed by the coating, likely through early activation of enzymes and enhanced hormonal signaling pathways.

Another critical consideration is the physiological and biochemical basis of priming, which has been explored in various crops. Several studies indicate that priming leads to partial advancement of the germination program without radicle emergence [61,62]. This includes the synthesis of mRNAs, the repair of damaged macromolecules, and the accumulation of stress-protective proteins. In rice, Chen and Arora [63] demonstrated that priming induced the expression of aquaporins and heat shock proteins, which contribute to better water regulation and stress tolerance during imbibition and early germination. These molecular changes likely contributed to the improved performance of primed seeds, especially under low-temperature and coated conditions.

Additionally, our findings have important agronomic implications. In direct-seeded rice systems, particularly in cooler climates or early spring sowing, the benefits of faster and more uniform germination can translate into better seedling establishment, reduced weed pressure, and improved yield potential. As shown by Meena et al. [64], seed priming improved yield and water-use efficiency in rice and wheat by ensuring robust early growth. Furthermore, the use of priming in conjunction with seed coating technologies could be optimized to deliver multiple benefits—nutrient supply through coatings and stress resilience through priming—provided the coating materials do not inhibit water uptake excessively.

While iron is an essential micronutrient that supports plant functions such as chlorophyll synthesis and respiration, its overuse can lead to toxicity. In excessive amounts, iron can disrupt nutrient balance, damage plant tissues, and ultimately hinder growth and development. This concern becomes especially important when considering sustained agricultural practices over multiple growing seasons.

A comparative analysis across these three years, measuring not only yield but also soil and plant tissue health, would provide crucial insights. It would help determine whether iron treatment is a sustainable solution or a short-term fix with long-term drawbacks. This kind of real-world, longitudinal study is vital for developing responsible agricultural practices, especially in regions where resources are limited and the margin for error is small. However, iron coating is recommended for direct sowing in developing countries with a lack of human resources. The iron-coated seeds could be directly sown with drones to reduce labor demands and ensure uniform seed distribution. This approach not only increases planting efficiency but also enables timely sowing over large areas, which is especially beneficial in regions facing agricultural labor shortages.

High temperature and humidity are among the most critical factors that negatively impact seed viability and longevity during storage. Exposure to elevated temperatures accelerates the metabolic activities within seeds, leading to faster depletion of energy reserves and increased production of reactive oxygen species (ROS), which in turn damage cellular structures and genetic material. Similarly, high humidity levels facilitate moisture absorption by seeds, creating an ideal environment for fungal growth and microbial activity, which further degrade seed quality. Together, these conditions promote rapid seed aging, resulting in a significant decline in germination rate, seedling vigor, and overall physiological performance. Maintaining low and stable temperature and humidity levels is therefore essential to preserve seed quality and ensure successful crop establishment.

Finally, our multi-year dataset enhances the reliability and generalizability of these findings. The consistency of priming benefits across varying seed lots and environmental conditions strengthens the argument for integrating priming into standard seed preparation protocols for rice cultivation. Future research should investigate the comparative effects of different priming methods, such as osmo-priming, hormo-priming, and biopriming, on both coated and non-coated seeds. Furthermore, transcriptomic and proteomic analyses could elucidate the underlying molecular mechanisms responsible for priming-induced improvements, particularly in aged seeds or under abiotic stress.

## 4. Materials and Methods

### 4.1. Seed Preparation and Seed Iron Coating

Rice seeds (*Oryza sativa* L. cv. Koshihikari) were provided by Ohmi Steel Industry company, which had been stored from three consecutive years (2022, 2023, and 2024) and were naturally aged; these seeds were stored at ambient room temperature during 2022 and 2023 without the application of standard storage conditions such as refrigeration or controlled light exposure. This method was intentionally selected to reflect the typical storage practices in many regions, where farmers often lack access to specialized facilities for maintaining optimal seed storage conditions. By simulating real-world scenarios, this study aimed to assess seed performance under practical, resource-limited conditions commonly faced in agricultural communities. Seeds were selected and thoroughly cleaned to remove dust, broken grains, and other impurities. Only healthy, uniform seeds with a moisture content below 12% were used for the coating process to ensure good adhesion and storability. Iron powder (S91 Premix, Ohmi Steel Industry, Osaaka, Japan) and calcined gypsum (CaSO_4_·0.5H_2_O) were used as the primary coating materials. A binder solution was prepared using 2% starch dissolved in warm distilled water, which was cooled to room temperature before application. The coating process was carried out using a manual mixing method. For every 1 kg of rice seed, 15 g of iron powder and 40 g of calcined gypsum were used. The rice seeds were placed in a clean, dry plastic drum and gently mixed with the starch binder until the seed surface became uniformly moist and tacky. Subsequently, the iron powder and calcined gypsum were gradually sprinkled over the seeds while continuously stirring to ensure uniform distribution and adherence to the seed surface. After coating, the seeds were spread in a single layer on clean trays and air-dried in the shade at ambient room temperature (25–30 °C) for 24 to 48 h. Once thoroughly dried, the coated seeds were stored in breathable paper bags under cool, dry conditions until further use (Figure 11).

### 4.2. Temperature Treatments

Seed germination and seedling growth were evaluated under seven temperature regimes—T1 (13 °C), T2 (15 °C), T3 (17 °C), T4 (19 °C), T5 (21 °C), T6 (23 °C), and T7 (25 °C)—using a controlled growth chamber (CHF-405 Cultivation Chamber, Tokyo, Japan). Germination was monitored over a period of 12 days under 10 h of light and 14 h of dark, while seedling growth was assessed up to 1 month after germination under the specified temperature conditions.

### 4.3. Seed Priming Treatment

Polyethylene glycol (PEG6000) (Cica-Reagent, Tokyo, Japan) [58] was used to prime the seeds before cultivation, and the seeds were soaked in the solution for 48 h. The seeds were then thoroughly dried by being washed 3 times with clean water and then left on a table for 3 h. The seeds were weighed after drying to approximate their initial weight. This experiment involved two priming treatments, −0 MPa and −1.5 MPa, based on the method used in previous studies [65]. To determine how much PEG6000 should be dissolved in pure water to create priming treatments, the following formula was employed:(1)$$OP=\left(-1.18\times 10^{-2}\right)\times C-\left(1.18\times 10^{-4}\right)\times C+\left(2.67\times 10^{-4}\right)\times C\times T+\left(8.39\times 10^{-7}\right)\times C^{2}T$$

In Equation (1), OP stands for osmotic pressure, the temperature is indicated by T, and PEG concentration is represented by C.

To prevent a decline in seed viability, the primed seeds were sown immediately after the priming process, without subjecting them to extended storage. This approach ensured that the physiological benefits of priming, such as enhanced germination and vigor, were fully retained at the time of sowing.

### 4.4. Seed Germination Parameters

Seed germination was counted daily, and other parameters such as seed relativized percentage (R%), mean germination time (MGT: day), mean germination rate (MGR: day^−1^), coefficient of variation of germination time ( CVt: %, seed day^−1^), coefficient of velocity of germination (CVG: %), germination index (GI: day), uncertainty of germination process (U: bit), synchronization index (Z), time to 50 % germination (T50: day), mean daily germination percent (MGD: %), peak value for germination (PV), and germination value (GV: day^−1^) were calculated using R software’s (version 4.1.2) germination package [66]. 

### 4.5. Seedling Height and Root Length

Seedling height and root length were measured 1 month after sowing using a standard ruler. For each treatment, measurements were taken from six individual plants, resulting in a total of 72 plants assessed across all treatments. Both seedling height and root length were recorded to evaluate early plant development and treatment effects. 

### 4.6. Statistical Analysis

We used R software (version 4.1.2) to analyze the data. To determine if there were significant differences between the treatment groups, we used a one-way analysis of variance (ANOVA), with a significance level set at *p* < 0.05. This means that we looked for differences in the results that were unlikely to have occurred by chance. 

## 5. Conclusions

This study comprehensively evaluated the effects of iron seed coating, cold temperature, and seed priming on the seed germination and early growth performance of rice seedlings. The results revealed that seeds subjected to iron coating alone showed delayed germination and lower vigor index compared to non-coated seeds, suggesting that high specific heat causes rapid temperature changes, or coating density may hinder oxygen and water uptake during germination. Furthermore, cold temperature stress (especially 13 °C) independently suppressed germination across all treatment groups, with marked reductions in germination percentage, germination speed, and seedling biomass. The combined stress of cold temperature and iron coating further intensified these negative effects, indicating a compounded inhibitory interaction that may disrupt metabolic activity during early seed development. In contrast, seed priming markedly improved germination performance, even under cold conditions. Primed seeds exhibited higher and faster germination, improved root and shoot elongation, and increased seedling vigor compared to non-primed in both coated and non-coated seeds. Based on these findings, it is concluded that while iron seed coating does not enhance and may in fact reduce germination, particularly under cold temperature stress, seed priming is an effective strategy to improve seed germination, vigor, and early growth of rice under both normal and adverse conditions. These results underscore the importance of optimizing seed enhancement techniques based on environmental context, where priming offers a more reliable approach for improving rice seed performance than iron coating.

## Figures and Tables

**Figure 1 plants-14-01683-f001:**
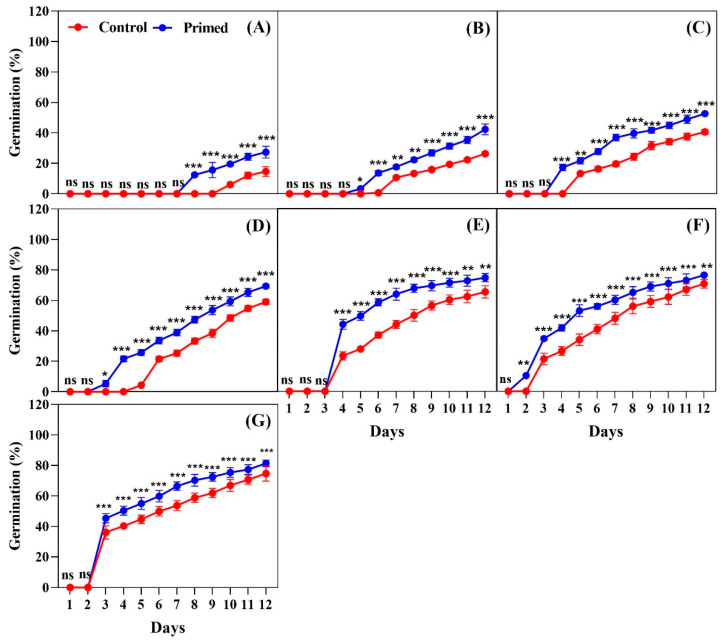
Effects of seed priming on the germination percentage of Koshihikari rice seeds harvested in 2022. Seed germination was tested at different temperatures: (**A**) 13 °C, (**B**) 15 °C, (**C**) 17 °C, (**D**) 19 °C, (**E**) 21 °C, (**F**) 23 °C, and (**G**) 25 °C. The values reflected in the graphs are the mean of 4 replications (n = 4), with each replication consisting of 100 seeds. Statistical significance is indicated as follows: ns (not significant), * *p* < 0.05, ** *p* < 0.01, and *** *p* < 0.001.

**Figure 2 plants-14-01683-f002:**
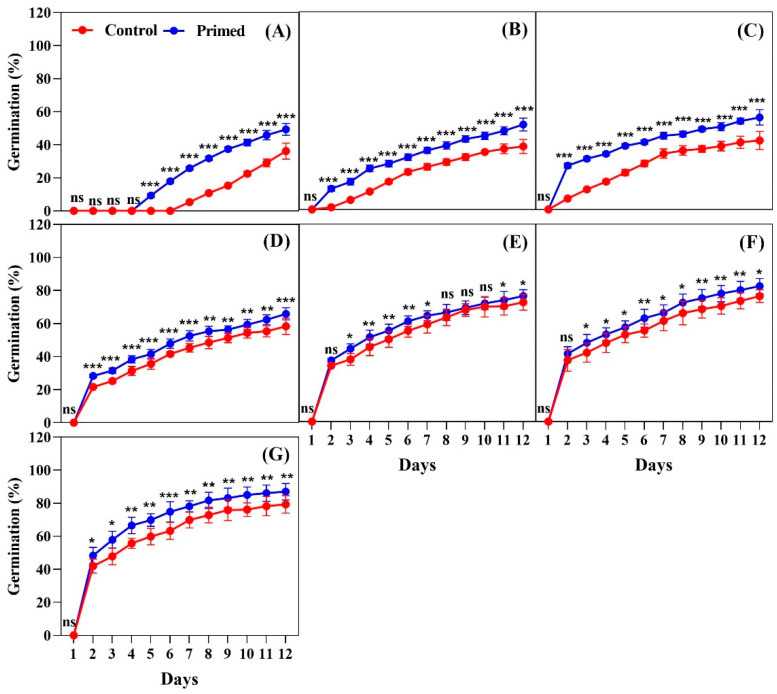
Effects of seed priming on the germination percentage of Koshihikari rice seeds harvested in 2023. Seed germination was tested at different temperatures: (**A**) 13 °C, (**B**) 15 °C, (**C**) 17 °C, (**D**) 19 °C, (**E**) 21 °C, (**F**) 23 °C, and (**G**) 25 °C. The values reflected in the graphs are the mean of 4 replications (n = 4), with each replication consisting of 100 seeds. Statistical significance is indicated as follows: ns (not significant), * *p* < 0.05, ** *p* < 0.01, and *** *p* < 0.001.

**Figure 3 plants-14-01683-f003:**
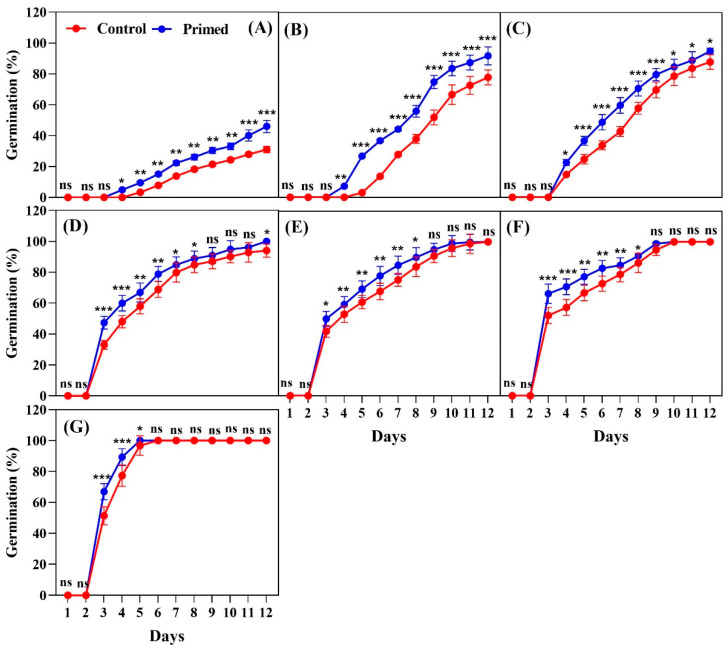
Effects of seed priming on the germination percentage of Koshihikari rice seeds harvested in 2024. Seed germination was tested at different temperatures: (**A**) 13 °C, (**B**) 15 °C, (**C**) 17 °C, (**D**) 19 °C, (**E**) 21 °C, (**F**) 23 °C, and (**G**) 25 °C. The values reflected in the graphs are the mean of 4 replications (n = 4), with each replication consisting of 100 seeds. Statistical significance is indicated as follows: ns (not significant), * *p* < 0.05, ** *p* < 0.01, and *** *p* < 0.001.

**Figure 4 plants-14-01683-f004:**
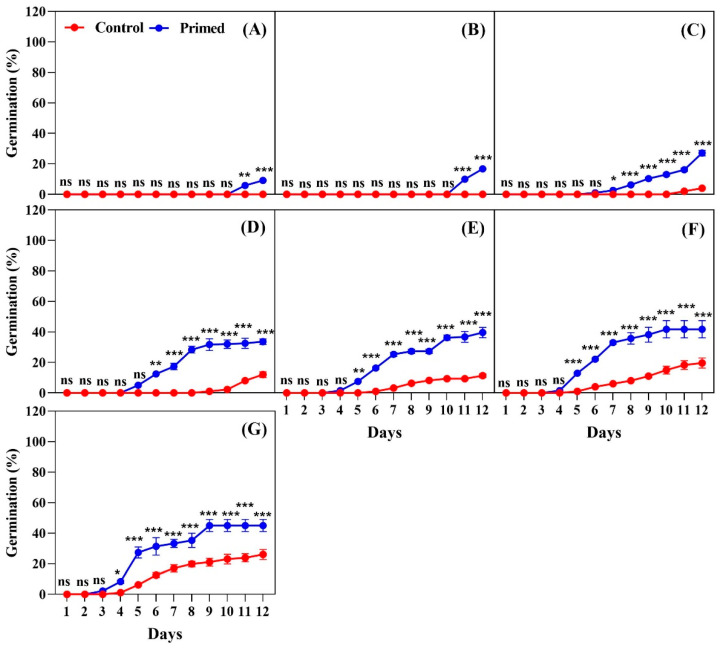
Effects of seed priming on the germination percentage of Koshihikari rice iron-coated seeds harvested in 2022. Seed germination was tested at different temperatures: (**A**) 13 °C, (**B**) 15 °C, (**C**) 17 °C, (**D**) 19 °C, (**E**) 21 °C, (**F**) 23 °C, and (**G**) 25 °C. The values reflected in the graphs are the mean of 4 replications (n = 4), with each replication consisting of 100 seeds. Statistical significance is indicated as follows: ns (not significant), * *p* < 0.05, ** *p* < 0.01, and *** *p* < 0.001.

**Figure 5 plants-14-01683-f005:**
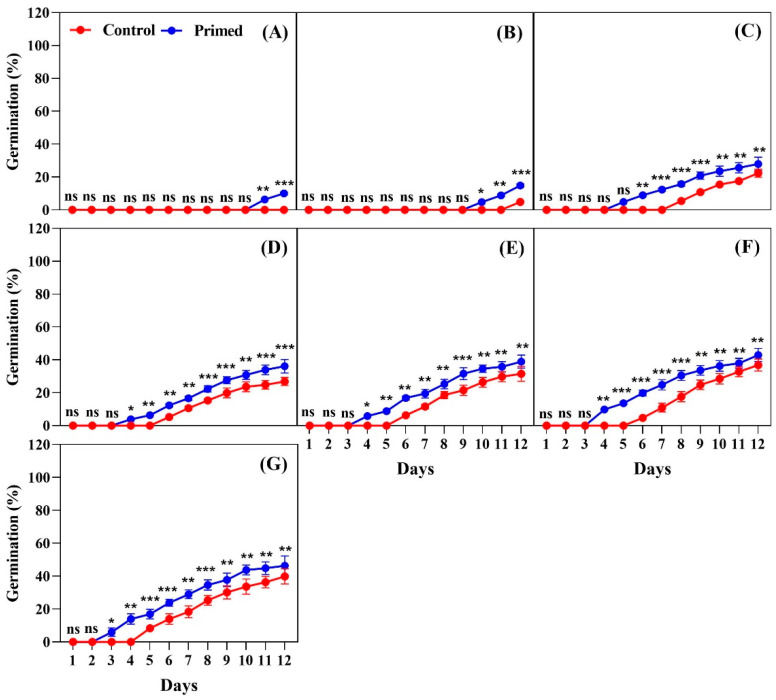
Effects of seed priming on the germination percentage of Koshihikari rice iron-coated seeds harvested in 2023. Seed germination was tested at different temperatures: (**A**) 13 °C, (**B**) 15 °C, (**C**) 17 °C, (**D**) 19 °C, (**E**) 21 °C, (**F**) 23 °C, and (**G**) 25 °C. The values reflected in the graphs are the mean of 4 replications (n = 4), with each replication consisting of 100 seeds. Statistical significance is indicated as follows: ns (not significant), * *p* < 0.05, ** *p* < 0.01, and *** *p* < 0.001.

**Figure 6 plants-14-01683-f006:**
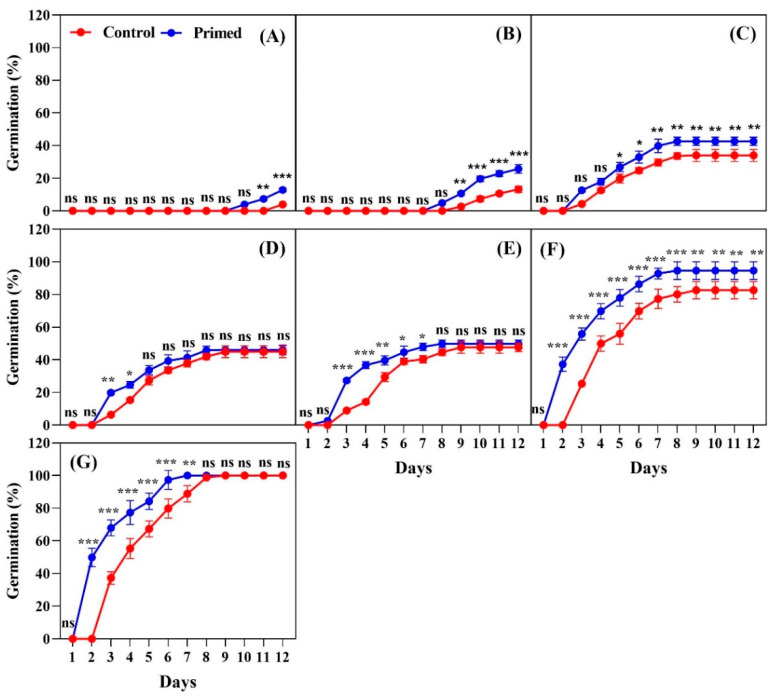
Effects of seed priming on the germination percentage of Koshihikari rice iron-coated seeds harvested in 2024. Seed germination was tested at different temperatures: (**A**) 13 °C, (**B**) 15 °C, (**C**) 17 °C, (**D**) 19 °C, (**E**) 21 °C, (**F**) 23 °C, and (**G**) 25 °C. The values reflected in the graphs are the mean of 4 replications (n = 4), with each replication consisting of 100 seeds. Statistical significance is indicated as follows: ns (not significant), * *p* < 0.05, ** *p* < 0.01, and *** *p* < 0.001.

**Figure 7 plants-14-01683-f007:**
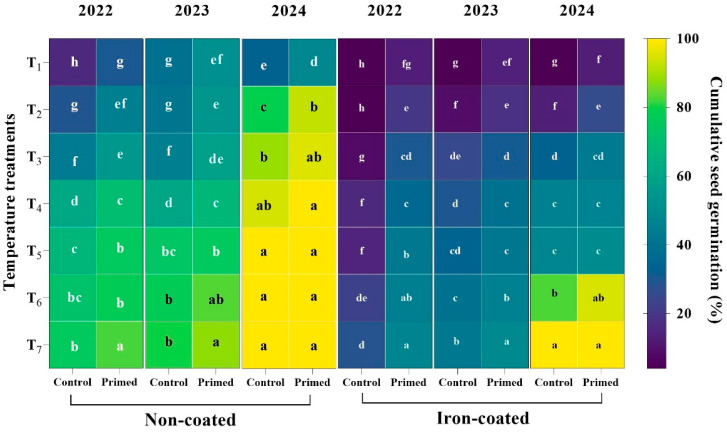
Effects of seed priming on the cumulative seed germination percentage of Koshihikari rice iron-coated seeds harvested in 2022, 2023, and 2024 at different temperature conditions. The values reflected in the graphs are the mean of 4 replications (n = 4), each replication consisting of 100 seeds. Different capital letters reveal the significant difference between temperatures, while different small letters reveal the significant differences between primed and control seeds. T1: 13 °C, T2: 15 °C, T3: 17 °C, T4: 19 °C, T5: 21 °C, T6: 23 °C, and T7: 25 °C.

**Figure 8 plants-14-01683-f008:**
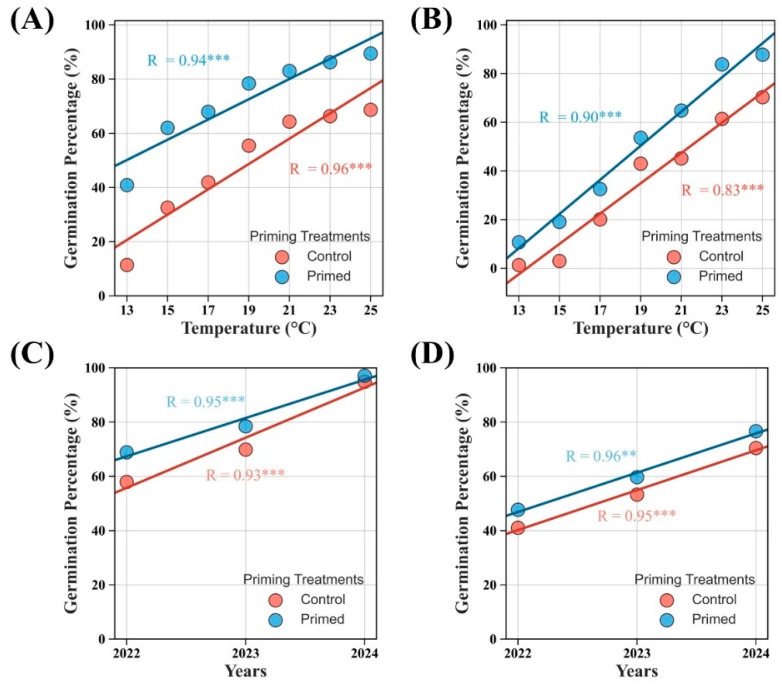
Effects of seed priming on the germination of non-coated (**A**) and coated (**B**) rice seeds at various temperature conditions. Panels (**C**,**D**) show the effects of seed priming on the germination of naturally aged non-coated (**C**) and iron-coated (**D**) rice seeds. ** *p* < 0.01, and *** *p* < 0.001.

**Figure 9 plants-14-01683-f009:**
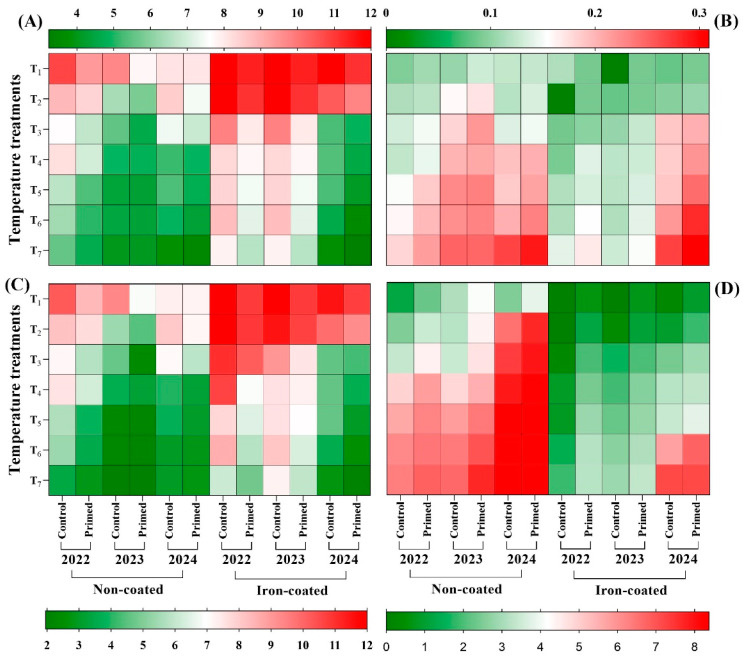
Effect of seed priming on mean germination time (**A**), mean germination rate (**B**), time to 50% germination (**C**), and mean daily germination (**D**) of control and primed seeds at different temperature conditions. Red color shows the highest value, while the green color reveals the lowest value of mean germination time, mean germination rate, time to 50% germination, and mean daily germination, respectively. The values reflected in the graphs are the mean of 4 replications (n = 4), with each replication consisting of 100 seeds. T1: 13 °C, T2: 15 °C, T3: 17 °C, T4: 19 °C, T5: 21 °C, T6: 23 °C, T7: 25 °C.

**Figure 10 plants-14-01683-f010:**
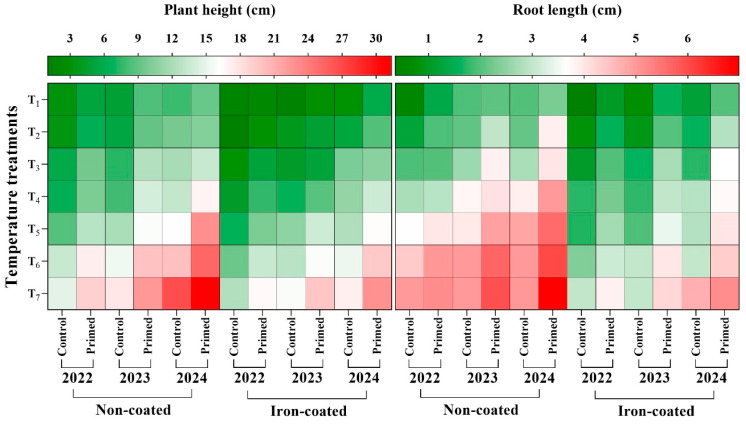
Effect of seed priming on plant height (**left**) and root length (**right**) of rice seedlings derived from non-coated and iron-coated seeds harvested in 2022, 2023, and 2024, grown at different temperature conditions. Red color shows the highest value, while the green color reveals the lowest value of plant height and root length, respectively. The values reflected in the graphs are the mean of 4 replications (n = 4), with each replication consisting of 100 seeds.

**Figure 11 plants-14-01683-f011:**
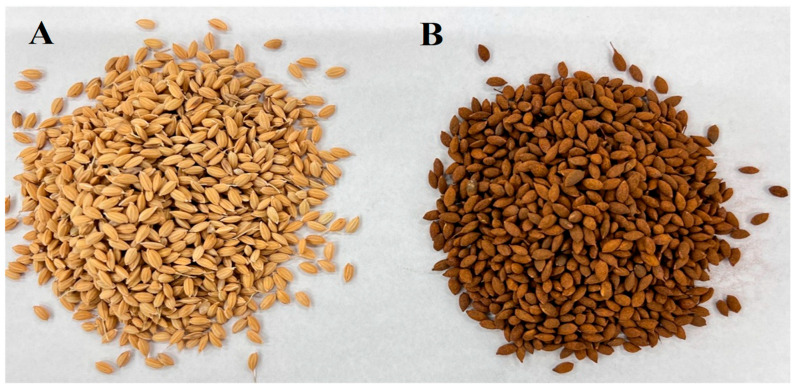
Seeds of the Koshihikari rice cultivar: (**A**) non-coated seeds and (**B**) iron-coated seeds.

## Data Availability

The raw data will be provided upon to request.

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
