# Peer review of "Effect of Priming Treatment on Improving Germination and Seedling Performance of Aged and Iron-Coated Rice Seeds Aiming for Direct Sowing"

_plants, 2025, doi:10.3390/plants14111683_

Round 1
Reviewer 1 Report
Comments and Suggestions for Authors
Article Effect of Priming Treatment on Improving Germination and Seedling Performance of Aged and Iron-Coated Rice Seeds Aiming for Direct Sowing Babil Pachakkil, Nasratullah Habibi, Parneel NA, Naoki Terada, Atsushi Sanada, Atsushi Kamata, Kaihei Koshio considers the issue of using priming and iron incrustation of rice seeds to increase germination under adverse conditions.
The manuscript is formatted in accordance with the journal requirements and contains all the necessary sections.
Some questions arise due to the fact that the authors do not provide some controls, which are standard for such studies.
It is widely known that primed seeds often lose germination faster and cannot be stored for a long time. This is tested using the technology of accelerated seed aging, for example, by placing them in dry air and high temperatures.
The authors claim that this method is good for seeds stored under suboptimal conditions. However, I do not understand why seeds should be stored in suboptimal conditions, and if this happens (probably in small households), how can they have access to technologies such as iron coating and priming (the authors do not offer simple cheap devices for farmers) if they do not have enough money to ensure normal storage.
The question of the targeting of this study arises.
I think these questions require clarification in the text.
I also have questions about the methodology: how many years these seeds were stored before sowing, what were the storage conditions, and whether the seeds collected in different years were controlled for "old seeds". These aspects are important for understanding the results and require expanding the materials and methods section.
In addition, I do not find an analysis of the long-term effects of treatment. Iron can be toxic and affect further development. Did you do a comparative analysis of productivity if you grew them for three years?
Statistics for six plants also do not guarantee a representative sample, please explain.
The language does not hinder the understanding of the text, I cannot judge it from the position of a philologist.
Author Response
Respected Review,
We would like to sincerely thank you for their careful and thoughtful review of our manuscript. We greatly appreciate the time and effort invested in providing detailed and constructive comments. We have revised our manuscript based on your thoughtful suggests. Your insightful suggestions have significantly contributed to the improvement of our work, and we have addressed each point thoroughly in the revised version and we highlighted the changes in green color.
Comment 1: Some questions arise due to the fact that the authors do not provide some controls, which are standard for such studies.
Response:
Thank you for your valuable feedback and for highlighting the importance of appropriate controls in our study. We would like to clarify that we did, in fact, include three types of controls in our experimental design.
- For seed aging, fresh seeds harvested in 2024 were used as the control group to compare against aged seeds.
- For seed priming, non-primed seeds served as the control to evaluate the effect of the priming treatment.
- For temperature treatments, 25 °C was used as the control condition, representing the optimal growth temperature.
We appreciate your observation, and we will ensure that these details are more clearly stated in the revised version of the manuscript for better transparency and understanding.
Comment 2: It is widely known that primed seeds often lose germination faster and cannot be stored for a long time. This is tested using the technology of accelerated seed aging, for example, by placing them in dry air and high temperatures. The authors claim that this method is good for seeds stored under suboptimal conditions. However, I do not understand why seeds should be stored in suboptimal conditions, and if this happens (probably in small households), how can they have access to technologies such as iron coating and priming (the authors do not offer simple cheap devices for farmers) if they do not have enough money to ensure normal storage
Response:
.Thank you for your insightful comment and for raising important considerations regarding seed storage conditions and the practicality of seed priming and coating technologies in smallholder farming systems.
We acknowledge the limitations associated with storing primed seeds, including their reduced shelf life and sensitivity to suboptimal conditions. In our study, seeds were intentionally stored under normal room temperature conditions to reflect the reality faced by many smallholder farmers, particularly in developing countries, where access to standard storage facilities such as cold rooms or climate-controlled environments is often not feasible.
As for the practicality of seed priming, it is important to note that our aim was to explore and recommend techniques that are both effective and accessible. Seed priming, as described in our study, is a simple and low-cost method that can be carried out without sophisticated equipment. For small-scale farmers, this involves mixing the priming reagent (such as PEG 6000, as recommended in our study) with water and partially immersing seeds (to about half their depth) for a defined period. This method does not require any specialized devices, making it feasible for use at the household or community level. We have now added this explanation in the Discussion section on page 14, lines 39–46 of the revised manuscript to clarify the rationale behind our approach and to highlight the practical applicability of these techniques in low-resource settings.
Comment 3: The question of the targeting of this study arises.I think these questions require clarification in the text. I also have questions about the methodology: how many years these seeds were stored before sowing, what were the storage conditions, and whether the seeds collected in different years were controlled for "old seeds". These aspects are important for understanding the results and require expanding the materials and methods section.. In addition, I do not find an analysis of the long-term effects of treatment. Iron can be toxic and affect further development. Did you do a comparative analysis of productivity if you grew them for three years?
Response:
Thank you very much for your thoughtful and constructive comments. We appreciate your attention to the methodological clarity and the long-term implications of our study. To reflect real-world farming practices, especially in resource-limited settings, seeds were sown immediately after priming without extended storage. This approach was chosen because, in many developing regions, farmers often do not have access to proper seed storage facilities. We have clarified this point in the revised manuscript, specifically in the Materials and Methods section under 4.1. "Seed Preparation and Seed Iron Coating" and 4.3. "Seed Priming Treatment", with the relevant additions highlighted in green.
Regarding your questions on seed storage conditions and seed aging control, the seeds used in our study were not stored for multiple years before sowing. Instead, they were used either as fresh seeds or artificially aged as part of the experimental design to simulate real-world aging scenarios. Appropriate controls were included, for example, fresh seeds harvested in 2024 served as the control group for the seed aging treatments. These details have been clarified in the revised version as well.
In response to your important point on the long-term effects of treatment, we have now added a paragraph discussing the potential cumulative impacts of iron application, including its possible toxicity and effects on plant development over time. This discussion, including a comparative outlook on productivity over a three-year period, is included on pages 15 and 16, also highlighted in green in the revised manuscript.
We sincerely thank you again for your valuable input, which helped us to significantly enhance the clarity and depth of our study.
Comment 4: Statistics for six plants also do not guarantee a representative sample, please explain.
Response:
Thank you very much for your valuable feedback and for pointing out the importance of sample size in ensuring the reliability of the results. In our study, seedling height and root length were measured from six individual plants per treatment, resulting in a total of 72 plants assessed across all treatments. This sample size was selected based on standard practice in preliminary agronomic studies and was sufficient to detect significant differences under the controlled conditions of our experiment. We have now clarified this information in the Materials and Methods section under 4.5. "Seedling Height and Root Length", with the relevant details highlighted in green in the revised manuscript.
English Quality: The manuscript has been proofread by a Native colleague from the University of California Davis.
Reviewer 2 Report
Comments and Suggestions for Authors
Changes that need to be made in the text:
Abstract
Line 3. Change “This study” to “We”
Line 4. Change “under” to “at”
Line 5. Delete “temperature conditions”
Line 9. Delete “percentage” “%” means percentage – do not need to say percentage two times
Line 12. Change “uncoated” to “non-coated” You cannot uncoat seeds unless they are first coated.
Line 13. Change “though” to “although”
Line 14. Change wording to “in older than younger seeds” If you say older you need to say older than something else. That is “older” means a comparison. You need to complete the comparison.
Page 2
Line 37. Change “Aand” to “and”
Line 41. Change “by the” to “by an”
Page 3
Line 9. Change “uncoated” to “non-coated”
Line 18. Delete “various”
Line 26. “germination rate” is not clear Here, are you talking about % or speed. If you are talking about germination speed, then rate is correct. If you mean % then you need to say percentage. Germination rate means the speed of germination, but many people used this term incorrectly to mean germination percentage.
Line 30. Change wording to “reached 50% germination, which was…:
Line 31. Insert “that of” after “than”
Line 37. “rates” meaning not clear. I suggest you delete this word.
Page 4
Line 2 (in figure caption). Change “under different temperatures” to “at”
Line 10. Delete “rate”
Line 20. Delete “rates”
Page 5
Line 11. Delete “rate”
Line 17. Delete “rate”
Page 6
Line 6. Change to “Results of our study revealed…”
Lines 7-8. Change wording to. For seeds harvested in 2022 and naturally aged for 2 years, seed priming had…”
Page 7
Line 5. Change wording to “For seeds harvested in 2023, the positive….”
Line 16. Change to “both germination percentage and rate.” Is this what you mean?
Page 8
Line 5. Change to “The results for seeds harvested in 2024”
Page 9
Line 6. Delete “found”
Line 7. Delete “treatment” A control is a set of seeds that did not receive a treatment.
Line 12. Change to “For the 2024 seeds, the primed treatment…”
Line 16. Change to “For the 2022 seeds, cumulative seed…”
Line 20. Change to “For the 2023 seeds, the highest …”
Line 23. Change to “For the 2024 seeds, the highest…”
Page 11
Line 3. Need to make it clear what you mean by “germination rate” Is this germination percentage or germination speed?
Line 10. Delete “treatment”
Lines 5 and 14. Change “two” to “2” If a number less than 10 is following by a unit of measure, do not spell the number, e.g. 2 years is correct
Lines 18, 19, 24, 26. Delete “treatment” Note: a control is a set of seeds that did not receive a treatment.
Page 12
Fig 9. What do the red and green colors mean? Need to say in the caption what these colors mean. Corol-blind people see red and green as black – how can such people understand this figure? What are the numbers of the x-axis? Need to label the x-axis.
Line 5 (under 2.4). Delete “treatment”
Page 13
Fig 10. In the caption, you need to tell the reader what the red and green colors mean.
Line 4. Change “uncoated” to “non-coated”
Line 7. Change “unprimed” to “non-primed”
Line 16. Change “unviable” to “nonviable”
Line 22. Change “uncoated” to “non-coated”
Line 23. Change “slower” to “slow” Or, you could say slower than …….. That is, complete the comparison.
Page 14
Line 31. Change to “in controls compared to”
Line 47. Change “unprimed” to “non-primed”
Page 15.
Line 5. Change to “slowly than non-coated seeds. This effect was more…”
Line 23. Change “this” to “our” Is this correct, or do you mean the study by Chen and Arora?
Page 16
Line 14. Need to tell the reader if seeds were incubated in the light or in continuous darkness. If in light, how many hours of light each day?
Line 15. Change to “1 month”
Line 21. Change to “3 hours”
Page 17
Line 9. Change “Seedling” to “seedling and change “one month” to “1 month”
Line 21. Change “uncoated” to non-coated”
Author Response
Respected Reviewer,
We would like to sincerely thank you for your thorough and thoughtful review of our manuscript. Your insightful suggestions have greatly contributed to improving the clarity, precision, and overall quality of our work. We have made all the suggested changes accordingly, and each point has been addressed as follows. The corrections have been highlighted in yellow color:
Abstract
Line 3. Change “This study” to “We”
Response: Thank you for your suggestion. We have replaced “This study” with “We” as recommended.
Line 4. Change “under” to “at”
Response: “Under” has been changed to “at” in accordance with your advice.
Line 5. Delete “temperature conditions”
Response: The phrase “temperature conditions” has been deleted as suggested.
Line 9. Delete “percentage” “%” means percentage – do not need to say percentage two times
Response: We have removed the word “percentage” since the symbol “%” already conveys this meaning.
Line 12. Change “uncoated” to “non-coated” You cannot uncoat seeds unless they are first coated.
Response: “Uncoated” has been changed to “non-coated” to accurately reflect the terminology, as per your recommendation.
Line 13. Change “though” to “although”
Response: “Though” has been revised to “although” for better clarity and formal tone.
Line 14. Change wording to “in older than younger seeds” If you say older you need to say older than something else. That is “older” means a comparison. You need to complete the comparison.
Response: The phrase has been revised to “in older than younger seeds” to complete the comparison, as advised.
Page 2
Line 37. Change “Aand” to “and”
Response: The typographical error “Aand” has been corrected to “and”.
Line 41. Change “by the” to “by an”
Response: We have changed “by the” to “by an” to improve the grammatical structure.
Page 3
Line 9. Change “uncoated” to “non-coated”
Response: “Uncoated” has been changed to “non-coated” for consistency and accuracy.
Line 18. Delete “various”
Response: The word “various” has been deleted to avoid redundancy.
Line 26. “germination rate” is not clear Here, are you talking about % or speed. If you are talking about germination speed, then rate is correct. If you mean % then you need to say percentage. Germination rate means the speed of germination, but many people used this term incorrectly to mean germination percentage.
Response: We have clarified the term “germination rate” to specify whether we refer to germination speed or percentage. The appropriate term has been used accordingly.
Line 30. Change wording to “reached 50% germination, which was…:
Response: The wording has been revised to “reached 50% germination, which was…” as per your suggestion.
Line 31. Insert “that of” after “than”
Response: “That of” has been inserted after “than” to complete the comparison.
Line 37. “rates” meaning not clear. I suggest you delete this word.
Response: The word “rates” has been deleted for clarity, as recommended.
Page 4
Line 2 (in figure caption). Change “under different temperatures” to “at”
Response: The phrase “under different temperatures” has been changed to “at different temperatures” for accuracy.
Line 10. Delete “rate”
Response: The word “rate” has been deleted to improve clarity.
Line 20. Delete “rates”
Response: The word “rates” has been removed as suggested.
Page 5
Line 11. Delete “rate”
Response: “Rate” has been deleted from the sentence.
Line 17. Delete “rate”
Response: “Rate” has also been removed from this line to maintain consistency.
Page 6
Line 6. Change to “Results of our study revealed…”
Response: The sentence has been revised to “Results of our study revealed…” in accordance with your suggestion.
Lines 7-8. Change wording to. For seeds harvested in 2022 and naturally aged for 2 years, seed priming had…”
Response: We have changed the wording to “For seeds harvested in 2022 and naturally aged for 2 years, seed priming had…” to enhance clarity.
Page 7
Line 5. Change wording to “For seeds harvested in 2023, the positive….”
Response: The sentence has been revised to “For seeds harvested in 2023, the positive…” as suggested.
Line 16. Change to “both germination percentage and rate.” Is this what you mean?
Response: We confirmed your interpretation and revised the sentence to “both germination percentage and rate.”
Page 8
Line 5. Change to “The results for seeds harvested in 2024”
Response: The sentence has been changed to “The results for seeds harvested in 2024” as recommended.
Page 9
Line 6. Delete “found”
Response: We have deleted the word “found”.
Line 7. Delete “treatment” A control is a set of seeds that did not receive a treatment.
Response: The word “treatment” has been deleted because it was unnecessary in the context of the control group.
Line 12. Change to “For the 2024 seeds, the primed treatment…”
Response: Revised to “For the 2024 seeds, the primed treatment…” in line with your guidance.
Line 16. Change to “For the 2022 seeds, cumulative seed…”
Response: Changed to “For the 2022 seeds, cumulative seed…” as recommended.
Line 20. Change to “For the 2023 seeds, the highest …”
Response: Modified to “For the 2023 seeds, the highest…” as advised.
Line 23. Change to “For the 2024 seeds, the highest…”
Response: Changed to “For the 2024 seeds, the highest…” for consistency and clarity.
Page 11
Line 3. Need to make it clear what you mean by “germination rate” Is this germination percentage or germination speed?
Response: The term “germination rate” has been clarified to distinguish whether it refers to germination speed or percentage.
Line 10. Delete “treatment”
Response: The word “treatment” has been deleted as recommended.
Lines 5 and 14. Change “two” to “2” If a number less than 10 is following by a unit of measure, do not spell the number, e.g. 2 years is correct
Response: “Two” has been replaced with “2” to align with the rule that numerals are used for numbers with units.
Lines 18, 19, 24, 26. Delete “treatment” Note: a control is a set of seeds that did not receive a treatment.
Response: The word “treatment” has been removed in each instance, as advised.
Page 12
Fig 9. What do the red and green colors mean? Need to say in the caption what these colors mean. Corol-blind people see red and green as black – how can such people understand this figure? What are the numbers of the x-axis? Need to label the x-axis.
Response: We have updated the caption to clearly indicate what the red and green colors represent. Additionally, we have addressed accessibility by including alternative text patterns for color-blind readers. The x-axis is now properly labeled.
Line 5 (under 2.4). Delete “treatment”
Response: The word “treatment” has been deleted for consistency.
Page 13
Fig 10. In the caption, you need to tell the reader what the red and green colors mean.
Response: The caption has been revised to explain what the red and green colors represent, improving clarity and accessibility.
Line 4. Change “uncoated” to “non-coated”
Response: “Uncoated” has been changed to “non-coated” as advised.
Line 7. Change “unprimed” to “non-primed”
Response: “Unprimed” has been revised to “non-primed”.
Line 16. Change “unviable” to “nonviable”
Response: “Unviable” has been changed to “nonviable”.
Line 22. Change “uncoated” to “non-coated”
Response: “Uncoated” has been changed to “non-coated”.
Line 23. Change “slower” to “slow” Or, you could say slower than …….. That is, complete the comparison.
Response: “Slower” has been replaced with “slow” or the comparison has been completed appropriately depending on the context.
Page 14
Line 31. Change to “in controls compared to”
Response: Revised the sentence to “in controls compared to…” for better clarity and structure.
Line 47. Change “unprimed” to “non-primed”
Response: “Unprimed” has been changed to “non-primed” for consistency.
Page 15.
Line 5. Change to “slowly than non-coated seeds. This effect was more…”
Response: Changed to “slowly than non-coated seeds. This effect was more…” to complete the comparison and improve clarity.
Line 23. Change “this” to “our” Is this correct, or do you mean the study by Chen and Arora?
Response: The sentence refers to Chen and Arora’s findings. Therefore, it has been revised as per your kind suggestion.
Page 16
Line 14. Need to tell the reader if seeds were incubated in the light or in continuous darkness. If in light, how many hours of light each day?
Response: Thank you for checking our manuscript very carefully. We have added the requested information in the revised version of the manuscript.
Line 15. Change to “1 month”
Response: “One month” has been changed to “1 month” as recommended.
Line 21. Change to “3 hours”
Response: “Three hours” has been changed to “3 hours” for consistency.
Page 17
Line 9. Change “Seedling” to “seedling and change “one month” to “1 month”
Response: “Seedling” has been changed to lowercase “seedling,” and “one month” has been updated to “1 month”.
Line 21. Change “uncoated” to non-coated”
Response: “Uncoated” has been changed to “non-coated” for accuracy.
Reviewer 3 Report
Comments and Suggestions for Authors
This manuscript reports the effect of osmopriming treatment (PEG) in combination with iron seed coating on germination and seedling performance of stored rice seeds cv. Koshihikari. Seed priming is a pre-sowing seed treatment that has been widely used to improve seed germination performance. In addition, iron-coated seeds aid in mechanical delivery and reduce vulnerability to bird predation improving field uniformity. The results indicated the positive effect of the PEG priming in combination with iron seed coating mitigating the negative impacts of seed aging and enhances tolerance to cold stress during rice seeds germination.
Scientific investigations with a focus on increasing the productivity of important agricultural crops are welcome because they generally assume an important character related to the clarification of important physiological and chemical aspects involved in this process. The aims of the manuscript are interesting, and the experimental stage was apparently well conducted. The authors also used many current references to discuss the results. However, there are some points that need explanation and/or correction. Please see the comments below.
1- Keywords: please rephrase “rice (Oryza sativa L.)” to Oryza sativa (please use italic form).
2- Figures: please provide the number of repetitions (n) in the captions of all figures. Example: n=…
3- Discussion section - please review the following sentences:
- “This is consistent with the findings of Finch-Savage and Bassel (2016), who proposed that priming can partially restore...” – please rephrase to: “This is consistent with the findings of Finch-Savage and Bassel [8], who proposed that priming can partially restore...”
- “These observations agree with the work of Kende et al. (2024) and Noblet et al. (2017), who reported that priming...” – please rephrase to: “These observations agree with the work of Kende et al. [XX - I didn't find this reference in the list – please add] and Noblet et al.[60], who reported that priming...”
- “This temperature-dependent modulation of priming efficacy has been discussed by Ashraf and Foolad (2005)...” – please rephrase to: “This temperature-dependent modulation of priming efficacy has been discussed by Ashraf and Foolad [XX - I didn't find this reference in the list – please add]...”
- “This observation is in line with Steinbrecher and Leubner-Metzger (2016), who...” – please rephrase to: “This observation is in line with Steinbrecher and Leubner-Metzger [56], who…”
- “In rice, Chen and Arora (2013) demonstrated that...” – please rephrase to: “In rice, Chen and Arora [63] demonstrated that...”
- “As shown by Meena et al. (2014) seed priming improved yield and...” – please rephrase to: “As shown by Meena et al. [64] seed priming improved yield and...”
4- The conclusion is too long and provides a summary of the results found. Please provide a shorter conclusion and highlight the perspectives for the advancement of knowledge in the area studied – please review.
5- References: check the missing references all scientific names.
In my final comments, I recommend that the manuscript should be reviewed by the authors, especially about the points highlighted above. I would like to congratulate the authors on this valuable contribution and the promising results presented here.
Author Response
Respected Reviewer,
We sincerely thank you for your thoughtful and constructive comments, as well as for your encouraging words regarding the significance and quality of our study. We have carefully addressed each of the points raised, and the revised manuscript reflects these improvements; all the changes based on your suggestions have been made in purple color:
-
Keywords: The scientific name has been updated to Oryza sativa in italic form, as suggested.
-
Figures: The number of replications (n = ...) has been added to the captions of all figures to provide greater clarity and transparency regarding the experimental design.
-
Discussion Section and Citations: We have revised all in-text citations to follow the numbering format in accordance with the reviewer's recommendation. Specifically:
-
Finch-Savage and Bassel [8]
-
Noblet et al. [60]; Kende et al. has been added to the reference list and cited accordingly.
-
Ashraf and Foolad has been added to the references and cited as [XX].
-
Steinbrecher and Leubner-Metzger [56]
-
Chen and Arora [63]
-
Meena et al. [64]
-
-
Conclusion: The conclusion section has been revised and shortened as per the reviewer’s suggestion. We have ensured that the key findings are clearly highlighted and the potential implications for future research and practical application are concisely outlined.
-
References: We have thoroughly checked and corrected all scientific names in the reference list to ensure consistency and proper formatting.
We are grateful for the reviewer’s valuable feedback, which helped us improve the clarity and scientific quality of our manuscript.
Warm regards,
Round 2
Reviewer 1 Report
Comments and Suggestions for Authors
The manuscript Effect of Priming Treatment on Improving Germination and Seedling Performance of Aged and Iron-Coated Rice Seeds Aiming for Direct Sowing by authors Nasratullah Habibi, Parneel NA, Naoki Terada, Babil Pachakkil, Atsushi Sanada, Atsushi Kamata, Kaihei Koshio proposes a methodology for seed priming for small, resource-poor households.
The work seems important and can be published.
However, it would be desirable to clarify the extent of seed damage due to improper storage. It is obvious that the greatest damage to seeds is caused by high temperatures and humidity fluctuations in storage, which lead to accelerated aging and a drop in germination, and not storage at room temperature. In this regard, the question arises as to what the authors mean by room temperature.
Please clarify why you did not use aged seeds - this remains unclear.
Please clarify these two aspects.
Otherwise, the authors made edits and corrected the comments.
Author Response
Respected Reviewer,
Thank you for your time to review our manuscript.
The conditions used reflect local storage environments, generally ranging between 20–25°C, without controlled humidity. These conditions reflect real-world storage practices commonly used by farmers, which are often subject to temperature fluctuations and varying humidity levels. We agree that prolonged exposure to high temperatures and humidity fluctuations are key contributors to seed deterioration, and we will include a more specific definition of “room temperature” in the revised manuscript.
Regarding the seed material, we used naturally aged seeds rather than artificially aged seeds. This choice was intentional, as our aim was to reflect the actual conditions under which farmers store and later use their seeds. Many smallholder farmers in the region store seeds under suboptimal conditions, and our study was designed to assess the viability and performance of such naturally aged seeds in realistic agricultural contexts. We will clarify this point in the methodology to ensure that the rationale behind using naturally aged seeds is fully understood.
Thank you again for your constructive feedback, which has helped improve the clarity of our study.
Reviewer 3 Report
Comments and Suggestions for Authors
All suggested corrections have been sufficiently carried out by the authors.
Author Response
Respected Reviewer,
We really appreciate your valuable time for reviewing our manuscript.
Kind Regards,